# *Everything for the users, nothing by the users*: Lessons learnt from an heterogeneous data mapping languages user study

Herminio García-González

IT and Communications Service, University of Oviedo, Asturias, Spain
`garciaherminio@uniovi.es`

**Abstract.** Usability has been seen as a new requirement in the Semantic Web community. Heterogeneous data mapping languages enable users to create knowledge graphs from legacy datasets. Even though some of these tools claim to be user friendly, this is not empirically demonstrated. In this paper, we briefly describe our usability experiment [1] with these languages and from its results we envisage next actions and problems that should be tackled in the topic. Covering them should lead to a better adoption among users.

**Keywords:** Data mapping languages · Data integration · Semantic Web · Usability

## 1 Introduction

Recent achievements in data mapping topic enable users to define heterogeneous data sources integration in a declarative fashion instead of using *ad-hoc* solutions which redounds in a higher flexible and faster process [8]. This fact allows users to invest less time and resources while constructing a knowledge graph. Thus, the final goal is to ease users' workflow. In addition, some of the proposed languages claim to be user friendly (i.e., YARRRML [5] and ShExML [2]) or easy to learn by semantic web experts (i.e., SPARQL-Generate [6]). However, this quality should be quantified in order to establish proper comparisons.

Recent trends in the semantic web community have seen the necessity to understand users and put them in the center of our solutions, improving their productivity and taking care of their needs[1]. Moreover, this has also been highlighted in the Knowledge Graph Construction W3C Community Group[2]. Users or usability studies allow to understand users' problems as well as their difficulties, needs and perceptions. They are, therefore, a huge analysis tool when deciding future actions on a topic.

However, to the far of our knowledge, only a recent study [1] has tackled the topic of usability in heterogeneous data mapping languages. In this paper

---

[1] http://www.juansequeda.com/blog/2020/11/16/international-semantic-web-conference-iswc-2020-trip-report/

[2] https://kg-construct.github.io/tpac-web/#report

we describe our experiences from mentioned user evaluation, we briefly describe the followed methodology, interesting results for the community and how them should drive future actions in the topic.

## 2   Brief experiment description

The experiment was designed as a mixed-method approach, that is to say, involving a quantitative and a qualitative design and analysis. On one hand, the quantitive part (involving capture of behavioural and performance metrics like keystrokes, completeness percentage, elapsed time, etc.) allows for an objective and direct evaluation of users interaction with tools and their tasks achievements. On the other hand, the qualitative part (involving a Likert scale questionnaire for variables like perceived easiness of use, learnability, applicability, etc.) enable to gather users subjective impression. Using a mixed-method approach we can correlate both set of measures to have a better understanding on how the users interact with and perceive the tool.

The sample consisted of 20 students pursuing a MSc in Web Engineering (first course out of two). The experiment was hosted the final day of the semantic web subject in which the students were introduced to semantic technologies (RDF, SPARQL, Shape Expressions, etc...). Therefore, we can categorise the sample as first-time users with some background knowledge. So, our results and conclusions will be in line with this described profile.

The sample was randomly distributed in three groups, one per language, so previous knowledge background bias could be mitigated. The experiment was divided in two tasks. The first one consisted in creating a set of mapping rules from a given input and the expected output. The second one was to perform a small modification to the mapping rules created in the former task. Therefore, first task measured global usability whereas the second one measured modifiability of the assigned language.

## 3   Results & Highlights

In first task 17 students submitted results (ShExML, 7; YARRRML 4; SPARQL-Generate 4) and in second task only 7 students did so (ShExML 6; YARRRML 1). These total results reveal that SPARQL-Generate users had problems when reaching a working mapping and that YARRRML users found difficult to modify an existing set of mapping rules or that they invested too much time in the first task.

We performed a statistical analysis (cf. [1]) per variable for the three languages as well as a pair-wise comparison to see in which variables and among which languages there were differences.

**Task 1:** In quantitative analysis significant differences were found in elapsed seconds (particularly between ShExML and YARRRML), completeness percentage (between ShExML and SPARQL-Generate) and precision (between ShExML

and SPARQL-Generate). This comes to corroborate that YARRRML users invested much more time than ShExML users when finding working solutions. The difference in completeness percentage and precision between ShExML and SPARQL-Generate reveals that SPARQL-Generate users were not able to find working solutions. In qualitative analysis significant differences were found in general satisfaction (between ShExML and YARRRML), learnability (between ShExML and other both languages), mapping definition easiness (between ShExML and other both languages) and easiness of use (between ShExML and YARRRML). These qualitative results come to corroborate and complement quantitative ones, so difficulties in finding working solutions by SPARQL-Generate users are translated to a worse learnability and mapping definitions easiness impression. More time consumed to find working solutions by YARRRML users is translated to worse impression in the four variables causing a descent in general usability indicators (i.e., general satisfaction level and easiness of use).

**Task 2:** In task 2 no significant differences could be established due to very low sample sizes. Only one YARRRML user was able to submit a non-working solution whereas 6 ShExML users submitted a solution for this task. This could be caused by the extra time needed by YARRRML users wrt ShExML users. ShExML modifiability variable was rated with 5 points by 83% of ShExML users and with 3 points by the YARRRML single user. SPARQL-Generate users were unable to reach this task as they had problems finishing the first one.

These differences reveal that the design of SPARQL-Generate is having a bad effect on first-time users which found it hard to operate and learn. However, it would be interesting to discern which part of the languages are causing the differences between ShExML and YARRRML. As an hypothesis, we can explain them due to their different syntaxes because ShExML uses keywords which can make the language more self-explanatory and offers modularity in its iterators which reminds the well-known by developers object-oriented paradigm.

Bad results in some qualitative variables for the three languages reveal another interesting picture. They perceive that the languages design lead to commit some errors (error proneness), that the error reporting system was not useful to solve their errors (error reporting system) and that they do not see much applicability to these tools (applicability). As we analyse further in the following section these three aspects should be handled urgently by the community.

## 4 Actions to take

In the light of the previously commented results it is important how new features are added and designed so they do not have a bad impact on users usability and learnability. In semantic web community we care a lot about new features and improvements but we tend to care less about users, we should involve them more.

Following the previous argument, users from our experiment told us that the three languages lead them to commit some errors (languages are not designed taking users mental models into account), that the error reporting system was not useful (it is another point that reveals that users are not involved in the

development process) and that they do not see much applicability in these tools. This last perception reveals something that could be extended to the whole semantic web community. With these languages and tools we are producing knowledge graphs, so in the end they do not see applicability to neither of them. In addition, applicability (and related variables like learnability) on first-time users reveal a derivate and correlated one: adoption. If first-time user do not see much applicability and technologies are hard to learn they are not going to adopt them. Therefore, these points are urgent ones that should be addressed by the community.

From a methodological point of view, we have to adopt stronger methods [9] which support our hypothesis and claims. Focusing in this community, Heyvaert et al. [4] compared user performance and perception from expert and non-experts users while using RMLEditor. In addition, they established a comparison with RML on users which had previous knowledge of it. However, comparisons are merely established using percentages. Lefrançois et al. [7] carried out a performance evaluation between RML and SPARQL-Generate main implementations. Again, this comparison is established based only on mean times. These weak comparisons could drive to some erroneous conclusions as they lack the power of a statistical test. In addition, we are losing the evidence strength measure of these conclusions. Thus, we have to learn from experimental[3] and social sciences on how observational experiments are performed, analysed and reported[4].

As we mentioned, our study only covered first-time users with some background knowledge so it is necessary to run these kinds of experiments with other profiles to have a whole perspective on the topic [9]. Doing so, we will be able to cover requirements from all users types and, thus, increase overall adoption. In addition, a comparison between visual and non-visual approaches along different users profiles should be carried out to discern users preferences and the target type of user to which each tool should be mainly addressed.

Finally, in order to contrast our hypothesis about differences between ShExML and YARRRML it would involve running more complex experiments which could come closer to the users' mental model processes so we can understand which language constructions and syntax are better. One possibility is to use cognitive models and frameworks [3] which could deliver explanations to our prior empirical study.

## 5   Conclusions

Semantic Web community recent trend to focus on users, understand them, and take care of their needs has opened another perspective in the semantic technological stack which advocates to put the user in the center of our thoughts.

Although some heterogeneous data mapping languages claimed its user friendliness, it was not empirically supported. Thus, in this paper we have briefly

---

[3] https://slideshare.net/miriamfs/vision-track-october2020fernandezv5
[4] https://slideshare.net/tammavalentina/the-tao-of-knowledge-the-journey-vs-the-goal

described our usability experiment in the field of heterogeneous data mapping languages where ShExML demonstrated a better usability on first-time users.

We have also claimed actions that should be addressed to solve elucidated common problems, further experiments that cover more users' profiles, more complex experiments that could explain users' mental model processes and the use of stronger methodological instruments and metrics.

Dealing with the exposed points we envisage a promising future for the Knowledge Graph Construction community, decreasing their technologies complexity barriers, having more users being attracted, and in short, improving their adoption.

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
