# OpenReview forum: "Everything for the users, nothing by the users: Lessons learnt from an heterogeneous data mapping languages user study"
_eswc-conferences.org/ESWC/2021/Workshop/KGCW — KGCW 2021_

### Official Review · ~Samaneh_Jozashoori1 · 2021-04-08
**Everything for the users, nothing by the users: Lessons learnt from an heterogeneous data mapping languages user study**

**Rating:** 6
**Confidence:** 3

**Review:**

This paper describes an experiment on the usability of three different data mapping languages based on a study on 20 first-time users. It opens the discussion on current existing problems in usability evaluation to be tackled and required actions to be taken.

The paper targets an interesting and challenging issue in the community, so it is well-suited to this workshop.

It is well-written overall and the contribution is well-described. However, I have some concerns:
* Although the **introduction** is easy to be followed by an expert in the domain, it is read more like a report than background knowledge and related work that is expected from a paper (since there is no other session for background or related work, the introduction is expected to provide at least a brief explanation of data mapping languages,...).
* It is explained that the user study is designed as a "mixed-approach" i.e. including both **quantitative** and **qualitative** analysis. As the results of the study are described in session 3, the qualitative analysis is not necessarily based on the variables other than the ones applied in quantitative analysis e.g. the time consumed to find working solutions which is a parameter in quantitative analysis, is also translated to "worse impression" as part of the qualitative analysis. So I am not quite convinced that the approach is a mixed- approach.

In summary, the paper provides a fair contribution for a workshop and can stimulate further user studies of this kind.

Minor typos:
* Title: "... from **a** heterogeneous..."
* page2, line3: "...how **they**"

---

### Official Review · ~Franck_Michel1 · 2021-04-08
**Interesting discusson on SW technos' usability - a few improvements and precisions needed**

**Rating:** 7
**Confidence:** 5

**Review:**

This paper summarizes a study reported in a previous paper, that seeks the evaluation of 3 languages designed for the mapping of heterogeneous data to RDF, from the view point of first-time users. It specifically focuses on how much users find the languages easy to learn and to use, and discusses the potential for adoption of these languages.
Then the paper suggests a set of actions that the SW community should pursue to improve the situation and lead to better adoption of these languages.

The paper is well written and easy to read. The conclusions made stem from the results of the study previously reported in [1], such that the reader has to trust the authors or go and read the previous paper.

Overall the paper points to an important issues that has been largely underestimated and disregarded by the community: the user's point of view. In this respect, the suggestions make sense and are worth spreading.

Nevertheless I'd like to stress some few issues that I think should be addressed in the camera ready version.


### Reference to the study paper
The paper should make much clearer from the very beginning that it is a summary of a previously published work.


### Focus on 1st-time users
The study focused specifically on 1st-time users, assuming that the first impression will decide of the adoption of the language. This is a bias that may lead to flawed conclusions.
Firstly, I would assume that users of these languages are likely to be recurrent users. In other words, they would not do a one-shot experiment and then never get back to it. In this context, having a demanding learning curve is not a problem since users will gain in expertise until they master the language.
Secondly, there are plenty of widely used computer languages that are hard to get a grip on. And still some of them are/were very successful. Meaning that the study should balance between user-perceived applicability and other reasons that make a language adopted or not.

### Choice of languages
Why only those 3 mapping languages? The only reason given is that they "clam to be user-friendly or easy to learn for SW experts". But because other mapping languages do not make that claim does not mean their are not user-friendly or easy to learn for SW experts. This needs better justification.
The original paper that is referred to ([1]) gives more explanation of this choice, but that should be made clearer here.

### Other misc. remarks

Experiment 2 is about making a "small modification" to the mapping made in exp 1. However, only 7 students responded as compared to 17 in exp 1. How come? Is that "small modification" hard enough to justify this?

Section 3 §1 says "SPARQL-Generate users had problems when reaching a working mapping" because no user gave results to the 2nd exp. But for YARRRML there was only one response and we learn later that it was wrong. So the conclusion on SPARQL-Generate seems really harsh here.

### Typos

- "to the far of our knowledge" -> to the best of our knowledge
- "interesting results for the community and how **them** should drive..." : them -> they
- "we can correlate both set of" : sets of
- "so they do not have a bad impact on **users** usability and learnability" : remove 'users'

---

### Official Review · ~Ahmad_Alobaid1 · 2021-04-20
**Review for the paper: lessons learnt from heterogeneous mapping languages**

**Rating:** 3
**Confidence:** 3

**Review:**

Summary. The author compares the usability of three mapping languages: YARRRML, ShExML, and SPARQL-Generate. The author focuses on first-time users. The experiment is done on 17 Master students and the author concluded that the usability of ShExML is better than the other two.

Good points:
The paper talks about an important topic and it is easy to read.

Major:
-	My main concern is the contribution of this paper in comparison to the previous study [1].
-	The experiment is briefly mentioned. I would expect to see more insights in comparison to the previously published one [1] (e.g., I would suggest including RML and R2RML).

Minor:
-	I like the idea of the quantitative comparison. However, the author mention that qualitative studies are not enough or “proper” for the usability. I would expect an argument or relevant references.
-	I prefer the abstract to be free from references.
-	I am missing a reference to the “Usability has been seen as a new requirement”.
-	The author uses a couple of URLs as references. In addition, I would prefer to see more proper references (a peer-reviewed article or something alike).
-	In page1, author mention their paper as the only paper which compare mapping languages. However, I prefer to see the author specifying that the reference is actually his previous work.
-	There are a couple of Typos (e.g., from an heterogeneous => from a heterogeneous).
-	In Section 3, line1-2, what does the numbers mean?
-	Why only 7 students did the second task?
-	The author described other experiments as could drive to erroneous conclusions as statistics are not used. I would expect more elaboration in this regard.

---

### Meta-Review · Program_Chairs · 2021-04-20

**Recommendation:** Accept
**Confidence:** 5

**Metareview:**

Dear authors,

This paper addresses an interesting and important topic for our community. But there are some important aspects of the paper that are lacking as mentioned by the reviewers:

- Inclusion of the required references, including the study paper
- Choice of the languages
- Detailed explanation of the study and its outcomes

It would be nice if these issues could be addressed in the camera-ready version of the paper.

---

### Decision · Program_Chairs · 2021-04-23

Accept